# Evaluation of Feline Exosome Mediated Renal Regeneration in Adenine-Induced Chronic Kidney Disease

**DOI:** 10.3390/biom15121647

**Published:** 2025-11-23

**Authors:** Chien Ming Lee, Brian Harvey Avanceña Villanueva, Hoang Minh, Qasim Hussain, Kuo Pin Chuang

**Affiliations:** 1Kao-Yi Animal Hospital, Chiayi 600, Taiwan; chase6383396@gmail.com; 2International Degree Program in Animal Vaccine Technology, International College, National Pingtung University of Science and Technology, Neipu Township, Pingtung 912, Taiwan; j11285355@mail.npust.edu.tw; 3Department of Anatomy and Histology, Faculty of Veterinary Medicine, Vietnam National University of Agriculture, Hanoi 100000, Vietnam; kira2108.hua@gmail.com; 4Department of Tropical Agriculture and International Cooperation, National Pingtung University of Science and Technology, Pingtung 912, Taiwan; 5Graduate Institute of Animal Vaccine Technology, College of Veterinary Medicine, National Pingtung University of Science and Technology, Neipu Township, Pingtung 912, Taiwan; 6School of Dentistry, Kaohsiung Medical University, Kaohsiung 807, Taiwan; 7School of Medicine, Kaohsiung Medical University, Kaohsiung 807, Taiwan; 8Companion Animal Research Center, National Pingtung University of Science and Technology, Neipu Township, Pingtung 912, Taiwan

**Keywords:** feline exosomes, chronic kidney disease, SD rats, adenine-induced nephropathy, regenerative therapeutics

## Abstract

Chronic kidney disease (CKD) is a progressive and irreversible condition that may lead to end-stage renal disease (ESRD). Current treatments can slow down the disease but cannot reverse kidney damage, creating a need for new regenerative therapies. Exosomes are small extracellular vesicles that show therapeutic potential by modulating inflammation, reducing cell death, and supporting tissue repair. This study established an adenine-induced chronic kidney disease (CKD) rat model through a two-week induction period. It resulted in significant weight loss, elevated serum creatinine (>1.3 mg/dL), and increased urinary protein levels (>30 mg/dL). After successful model establishment, exosome treatment was administered. During the 4-week treatment phase, both treatment and control groups showed gradual recovery in body weight. Serum creatinine slightly decreased but remained above the normal range, and urinary protein levels trended toward normalization. No functional improvements were clearly attributable to exosome treatment. However, histopathological analysis revealed that the exosome treated group exhibited marked structural improvements, including reduced renal degeneration, cyst formation, and tubular dilation. These findings indicate that while exosome therapy did not produce significant short-term functional recovery, it may confer structural protective effects in CKD.

## 1. Introduction

Chronic kidney disease is a progressive and irreversible decline in kidney function [1]. It contributes to increased cardiovascular events, end-stage renal disease (ESRD), and a diminished quality of life [2,3,4]. Current therapeutic strategies primarily focus on managing symptoms and slowing disease progression, rather than achieving true renal regeneration or reversal of damage [5,6]. These approaches, including strict blood pressure control [7], glycemic management, and dietary modifications [8], often fail to stop the gradual decline in kidney function. This ultimately leads to the need for renal replacement therapies (RRTs) such as dialysis or kidney transplantation. These limitations of existing treatments highlight an urgent need for novel therapeutic interventions that can actively promote kidney repair and regeneration.

In recent years, exosomes have emerged as a new frontier in regenerative medicine [9]. Exosomes are nano-sized lipid bilayer vesicles (30–150 nm), containing a diverse cargo of proteins, lipids, mRNA, and microRNAs [10]. They act as crucial mediators of intercellular communication, transferring their contents to recipient cells and influencing a wide array of physiological and pathological processes [11,12]. Growing evidence suggests that exosomes derived from various sources, including mesenchymal stem cells (MSCs), have potent regenerative, anti-inflammatory, and anti-fibrotic properties [13]. It makes them highly attractive candidates for treating kidney diseases [14]. Their ability to deliver bioactive molecules directly to injured renal cells, modulate immune responses, and promote tissue repair pathways positions them as a promising cell-free therapeutic strategy for renal regeneration [15].

The adenine-induced chronic kidney disease (CKD) rat model has gained significant recognition as a reliable and clinically relevant experimental model for studying the pathogenesis and therapeutic interventions for CKD [16]. This model effectively reproduces the characteristic features of human CKD pathophysiology, including progressive renal dysfunction, tubulointerstitial fibrosis, inflammation, and crystal formation [17]. It does so by inducing the accumulation of 2,8-dihydroxyadenine (DHA) crystals in the renal tubules [18]. It has an ease of induction, reproducibility, and the close resemblance of its pathological manifestations to human CKD. These characteristics make it an invaluable tool for preclinical research aimed at identifying novel reno-protective and reno-regenerative strategies [16]. Unlike surgical models, which introduce procedural variability [19], the adenine-induced model provides a standardized, non-invasive approach for consistent evaluation of therapeutic interventions.

Given the urgent need for effective CKD therapies and the promising regenerative potential of exosomes, this study aims to evaluate the efficacy of feline exosome-mediated renal regeneration in an adenine-induced CKD rat model. We hypothesize that administration of exosomes will reduce renal damage, promote structural repair, and improve kidney function. The findings from this study are expected to contribute significantly to the development of novel strategies for combating the devastating effects of chronic kidney disease (CKD).

## 2. Materials and Methods

### 2.1. Experimental Design

This study followed the animal study protocol reviewed and approved by the Institutional Animal Care and Use Committee (IACUC; protocol code: 25PLAN-21; approved on 24 February 2025). 30 male Sprague-Dawley (SD) rats of 6 weeks were used for this study. They were purchased from BioLASCO, Co., Ltd, Taipei, Taiwan. All animals were quarantined and acclimatized before treatment, and a veterinarian confirmed their healthy status. The animals were randomly divided into two groups, treatment group (*n* =15) which received weekly subcutaneous injections of exosomes for 4 weeks and control group (*n* = 15) which received weekly subcutaneous injections of normal saline for 4 weeks.

### 2.2. Feeding and Care

The rats were kept at an environment of 22 ± 3 °C with 30–70% humidity. The light cycle consists of 12 h of light and 12 h of dark. Two rats were kept per cage. The rats were fed with Lab Diet^®^ 5001 Rodent Diet (Lab Diet, Richmond, IN USA) *ad libitum* and were provided with RO water for drinking *ad libitum.*

### 2.3. Exosome Characterization

Exosomes derived from cat umbilical cord mesenchymal stem cells were produced through a standardized clinical manufacturing process. We began with culturing cat umbilical cord mesenchymal stem cells at a concentration of 1 × 10^8^ cells/mL. Subsequently, the cell culture supernatant was collected and underwent pH and conductivity optimization. The product was then concentrated 15-fold using tangential flow filtration. To preserve structural integrity, ADP was incorporated into the formulation, after which the material underwent lyophilization. The final sterilization step involved gamma irradiation.

### 2.4. Size Distribution Analysis

Nanoparticle tracking analysis (NTA) was performed using NanoSight NS300 (Malvern Panalytical, Malvern, Worcestershire, UK) to determine exosome size distribution and concentration. Sample preparation involved dilution in phosphate-buffered saline with thorough mixing. Analysis parameters included laser illumination at 532 nm wavelength and video capturing for 60 s at an average of 30 frames per second [20,21]. The vesicle population displayed a modal size of ~92 nm, consistent with the exosome size range (30–150 nm). The overall concentration was also on the order of 10^8^ particles/mL, within the recommended quantification range (see Figure 1 and Figure 2).

### 2.5. Chronic Kidney Disease (CKD) Induction

Each rat was weighed prior to the beginning of the induction period (tolerance within ±1 g). Adenine was suspended in 0.5% methylcellulose to prepare a dosing solution of 300 mg/kg body weight per day. The suspension was prepared fresh daily by vortexing to ensure even dispersion. Using an oral gavage needle and syringe, each rat was administered adenine suspension once daily for 14 consecutive days. Animals were monitored for general condition, body weight, and signs of distress. At the end of the 2-week induction period, CKD was confirmed based on reduced body weight, enhanced serum creatinine levels (determined through creatinine testing strips from IDEXX Laboratories, Inc, Westbrook, ME, USA. Model: Catalyst One) and increased urinary protein excretions (measured by AUTION Sticks 10EA; ARKRAY Factory, Inc., Shiga, Japan; Cat. No. 73591).

### 2.6. Exosome Therapeutic Treatment

Lyophilized exosome powder was reconstituted in 2 mL of sterile phosphate-buffered saline (PBS) prior to each administration. The solution was gently swirled until fully dissolved before use. Each rat received 0.1 mL of exosome suspension per injection, corresponding to 8.80 × 10^9^ ± 1.27 × 10^8^ particles per animal, based on the particle concentration of 8.80 × 10^10^ ± 1.27 × 10^9^ particles/mL (mode size ≈ 92 nm). Given the average body weight of 146 g, the effective dose was 6.0 × 10^10^ ± 0.9 × 10^9^ particles/kg body weight. Rats in the treatment group received weekly subcutaneous injections of exosomes for a total duration of 4 weeks. On the other hand, rats in the control group received weekly subcutaneous injections of 0.1 mL sterile phosphate-buffered saline (PBS), administered on the same schedule as the treatment group. Throughout the 4-week treatment period, body weight and general health status were monitored and recorded.

### 2.7. Animal Assessment and Sample Collection

During the study, body weight was measured throughout both the induction and treatment phases as an indicator to assess the animals’ overall health status and nutritional condition. In addition, body weight monitoring was helpful to observe the systemic responses of the animals to experimental procedures during model establishment and treatment. It also provides important reference data for subsequent analysis of therapeutic efficacy and safety. Serum creatinine and urinary protein levels were measured (at the baseline) before induction, post-induction, and post-treatment (at the end of the study). All animals were euthanized under anesthesia. The kidneys were exercised, fixed, and processed for histological sectioning to assess structural changes and pathological damage in the renal tissue.

### 2.8. Histopathological Analysis

Histopathological examination was conducted following standard histopathological method [22]. Morphological features were examined through hematoxylin and eosin (H&E) staining of tissue specimens [23], and collagen deposition was evaluated using Masson’s trichrome staining [24]. Microscopic observations were conducted with an Olympus BX53 light microscope (manufactured by Olympus Corporation, Tokyo, Japan; sourced from Yuan Yu Industrial Co., Ltd., Taipei, Taiwan). The criteria of the severity grading system for inflammation, tubular necrosis, fibrosis, degeneration, tubular dilation, and cyst lesions were based on [22], see Table 1.

### 2.9. Data Analysis and Writing

All experimental data were recorded in Microsoft Excel (2010, Microsoft Corporation, Redmond, WA, USA) and analyzed using R statistical software (Version 4.4.2, R Foundation for Statistical Computing, Vienna, Austria) integrated with RStudio (Version 2025.09.2, Posit, Boston, MA, USA). Data processing was performed using tidyverse packages, readxl, dplyr, tidyr, and stringr [25], while graphical representations were created using ggplot2, ggbeeswarm, cowplot, and scales packages. Longitudinal body weight data were analyzed using Linear Mixed-Effects Model (LMEM) with Group and Time as fixed factors, their interaction, and Animal ID as a random effect. Post-hoc pairwise comparisons were performed using estimated marginal means with Tukey adjustment for multiple comparisons. Serum creatinine levels were analyzed using LMEM to test the Group × Phase interaction, with Animal ID as a random effect. Given non-normal distributions, definitive comparisons used non-parametric tests: paired Wilcoxon Signed-Rank Test for within-group changes and Wilcoxon Rank-Sum Test on delta (Δ) scores for between-group differences. Pairwise comparisons were adjusted using Holm-Bonferroni correction. After confirming normality and homogeneity of variance, histopathological parameters were compared using unpaired Student’s *t*-test with Holm correction for multiple comparisons. Body weight data are presented as means with 95% confidence intervals using beeswarm plots; serum creatinine as mean ± SEM with individual points overlaid on bar graphs; and histological scores as mean ± SD. Statistical significance was set at α = 0.05. Furthermore, GenAI tool Claude Sonnet 4.5 was used for text editing and language improvement.

## 3. Results

### 3.1. Chronic Kidney Disease Induction

#### 3.1.1. Body Weight During Induction

After two weeks of induction, the average body weight decreased by approximately 40 g per rat (see Figure 3). The decrease in body weight was gradual.

#### 3.1.2. Serum Creatinine and Urinary Protein Levels After Induction

Serum creatinine concentrations in all animals exceeded 1.3 mg/dL (see Table 2 and Table 3 and Figure 4), indicating renal impairment. Furthermore, urinary protein levels were consistently greater than 30 mg/dL. Collectively, these findings confirm the successful establishment of the CKD model.

### 3.2. Exosome Therapeutic Treatment

#### 3.2.1. Body Weight After Exosome Treatment

During the treatment phase, weeks 3 to 6, both treatment and control groups exhibited a gradual recovery in body weight (see Figure 5). The statistical difference between control and treatment groups was not significant.

#### 3.2.2. Serum Creatinine and Urinary Protein Levels After Treatment

Serum creatinine levels were decreased (both in control and treatment groups) after the treatment, but there was no statistically significant difference between the groups. Overall, the serum creatinine levels (in both groups) remained above the normal range (see Table 4 and Figure 4). Urinary protein levels remained undetected.

### 3.3. Histological Analysis

Histopathological analysis (see Figure 6 and Figure 7), however, revealed that the treatment group demonstrated marked improvements as compared to the control group in renal degeneration, cyst formation, and tubular dilation.

### 3.4. Data Analysis

Longitudinal body weight analysis using Linear Mixed-Effects Model (LMEM) revealed significant main effects of Time during both induction (F = 831.32, *p* < 0.001) and treatment phases (F = 773.72, *p* < 0.001), but no significant Group effects (Induction: F = 0.22, *p* = 0.637; Treatment: F = 0.40, *p* = 0.526) or Group × Time interactions (Induction: F = 0.28, *p* = 0.755; Treatment: F = 0.36, *p* = 0.839), indicating comparable weight trajectories between Control and Treatment groups throughout both study phases. Post-hoc pairwise comparisons confirmed no significant between-group differences at any time point. For serum creatinine, LMEM analysis showed a significant main effect of Phase (F = 436.33, *p* < 0.001) but no significant Group × Phase interaction (F = 0.15, *p* = 0.865), indicating similar temporal trends between groups. Holm-corrected paired Wilcoxon Signed-Rank Tests revealed significant increases from baseline to post-induction in both Control (*p* = 0.002) and Treatment groups (*p* = 0.002), with no significant difference in change magnitude (Wilcoxon Rank-Sum on Δ scores: W = 108, *p* = 0.868). Comparing post-induction to post-treatment, creatinine levels significantly decreased in both Control (*p* = 0.002) and Treatment groups (*p* = 0.002), and the Δ score comparison again showed no significant between-group difference (W = 128.5, *p* = 0.517). Histopathological analysis using unpaired Student’s *t*-test demonstrated a significant reduction in total histological damage score in the Treatment group compared to Control (Control: 19.2 ± 1.78 vs. Treatment: 16.47 ± 1.46, t = 4.60, df = 28, *p* < 0.001, Cohen’s d = 1.68). After Holm correction for multiple comparisons, three individual parameters showed significant improvements: Cyst formation (*p* < 0.001, Cohen’s d = 1.93), Dilation (*p* = 0.002, Cohen’s d = 1.50), and Degeneration (*p* = 0.002, Cohen’s d = 1.45), while Fibrosis, Necrosis, and Inflammation did not reach significance after correction (all adjusted *p* > 0.20).

## 4. Discussion

This study investigates the therapeutic efficacy of exosome treatment in an adenine-induced chronic kidney disease (CKD) rat model, focusing on both structural and functional outcomes. Our findings demonstrate that while exosome treatment did not lead to significant short-term functional recovery, as evidenced by persistent elevated serum creatinine and urinary protein levels, it promoted structural amelioration in tubular injury parameters. Histopathological analysis revealed a significant reduction in renal degeneration, cyst formation, and tubular dilation. This structural restoration, without accompanied by immediate functional improvement, suggests a complex interplay between tissue repair mechanisms and overall organ function in chronic kidney disease. It suggests that the structural repair mediated by exosomes may precede the functional restoration in the context of adenine-induced CKD.

This phenomenon is not completely new in regenerative medicine. It is probable that the initial phase of exosome-mediated repair primarily focuses on restoring tissue architecture and cellular integrity, which are prerequisites for subsequent functional improvement. The adenine-induced CKD model is characterized by extensive tubular damage and interstitial fibrosis [26], and the observed reduction in tubular dilation and degeneration points towards a regenerative effect of exosomes on the damaged nephron segments. This is consistent with the recent studies demonstrating the pro-regenerative capacity of exosomes in the kidney. Exosomes derived from mesenchymal stem cells (MSCs), for instance, have been shown to promote renal tubular cell proliferation and survival, thereby facilitating the repair of damaged nephrons [27]. Exosomes are known to carry a diverse cargo of growth factors, cytokines, and genetic material (e.g., miRNAs) that may promote cell proliferation, differentiation, and angiogenesis, thereby contributing to tissue regeneration [28]. The cargo of exosomes, including growth factors like HGF and IGF-1, and microRNAs, may also activate pro-survival and pro-proliferative pathways in recipient cells, contributing to the observed structural improvements [29].

The adenine-induced CKD model is a well-established and clinically relevant model that mimics many of the key features of human CKD, particularly tubulointerstitial fibrosis and inflammation [30]. The administration of adenine leads to the formation of 2,8-dihydroxyadenine crystals in the renal tubules, causing obstruction, tubular cell injury, and a subsequent inflammatory and fibrotic response [18]. This model is particularly advantageous for studying therapies aimed at tubular regeneration, as the primary site of injury is the renal tubule. Our findings of reduced tubular dilation and degeneration following exosome treatment align well with the known pathology of this model and highlight the potential of exosomes to directly target and repair damaged tubular structures. However, the adenine model also presents some challenges. The persistent crystal deposition can act as a continuous stimulus for inflammation and fibrosis, making it difficult to achieve complete resolution of these pathological processes. This may explain why we did not observe significant improvements in fibrosis, inflammation, and necrosis, despite the clear evidence of tubular amelioration.

Our study has limitations that should be addressed in future investigations. First, longer-term studies are required to determine if the observed structural improvements eventually lead to significant functional recovery. Second, exploring combination therapies, such as exosomes with anti-fibrotic drugs or agents that target crystal deposition, could offer synergistic benefits. Third, we did nanoparticle tracking analysis (NTA) only for exosome characterization. Fourth, we could not assess the molecular markers (e.g., Ki67, KIM-1, NGAL, α-SMA, collagen-I, or TGF-β) and biochemical parameters (BUN, eGFR, urine microalbumin, and cystatin-C). Overall, our study provides evidence that exosome treatment in an adenine-induced CKD rat model promotes structural amelioration in tubular injury parameters, leading to significant improvements in renal degeneration, cyst formation, and tubular dilation. This amelioration capacity, however, did not translate into immediate functional recovery, nor did it significantly enhance fibrosis, inflammation, and necrosis within the 4-week treatment period.

These findings carry significant implications for the understanding and treatment of chronic kidney disease. The observed structural amelioration, even in the absence of immediate functional improvement, highlights the potential of exosome-based therapies to alter the progression of CKD by promoting tissue repair at a cellular level. This amelioration capacity could be particularly valuable in early stages of CKD, or as an adjunct therapy to prevent further deterioration and preserve residual renal function. The ability of exosomes to reduce renal degeneration, cyst formation, and tubular dilation can be harnessed to address the underlying pathological changes that drive CKD progression, moving beyond symptomatic management to genuine tissue restoration.

## 5. Conclusions

This study successfully established a chronic kidney disease (CKD) rat model, as evidenced by significant weight loss, elevated serum creatinine, and increased urinary protein levels following induction. While exosome treatment did not result in immediate improvements in functional parameters such as serum creatinine and urinary protein, histopathological evaluation revealed benefits, including reduced renal degeneration, cyst formation, and tubular dilation. These findings suggest that exosome therapy may confer structural protective effects in CKD, even in the absence of marked short-term functional recovery.

## Figures and Tables

**Figure 1 biomolecules-15-01647-f001:**
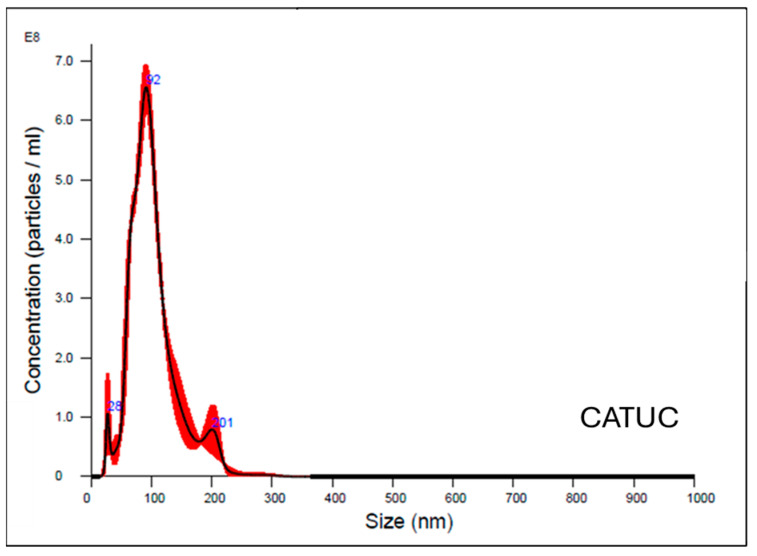
Nanoparticle tracking analysis of isolated exosomes. The size distribution shows a modal diameter of ~92 nm, consistent with the characteristic size range of exosomes. The black line denotes the mean particle concentration, and the red shading corresponds to the standard deviation.

**Figure 2 biomolecules-15-01647-f002:**
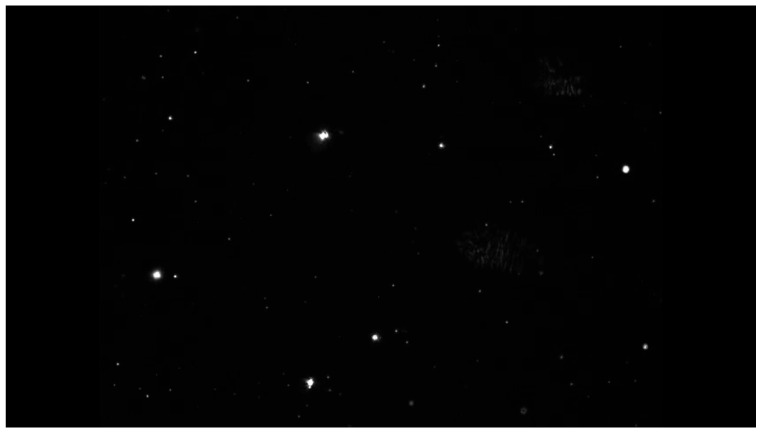
Nanoparticle tracking image shows individual exosome particles.

**Figure 3 biomolecules-15-01647-f003:**
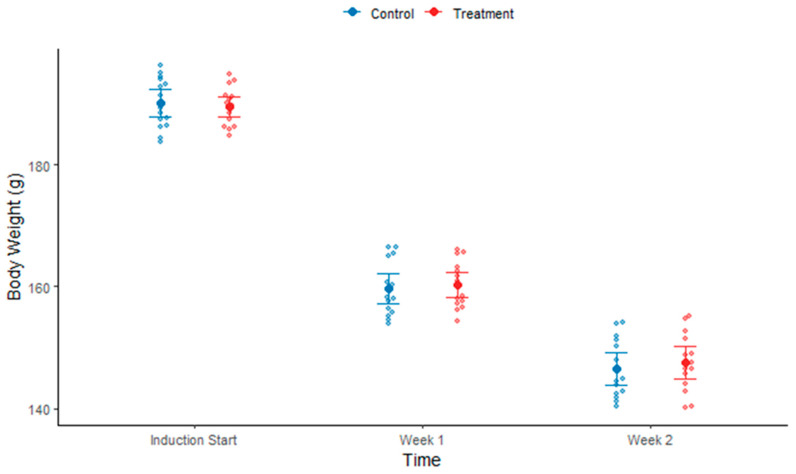
Bee swarm plot shows body weight reduction during the induction phase after week 1 and week 2.

**Figure 4 biomolecules-15-01647-f004:**
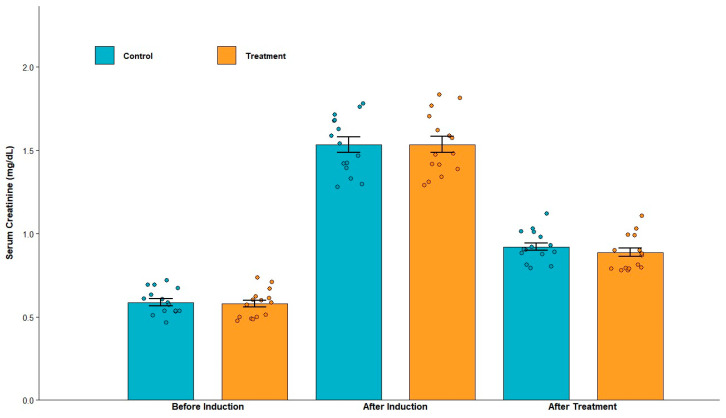
Serum creatine levels before chronic kidney disease (CKD) induction, after chronic kidney disease (CKD) induction, and after treatment. Small circles represent individual data points. After induction, serum creatinine levels are above the normal range. Serum creatinine levels after treatment showed a decrease compared with post-induction values; however, they remained above the normal range and there was no statistically significant difference between control and treatment groups.

**Figure 5 biomolecules-15-01647-f005:**
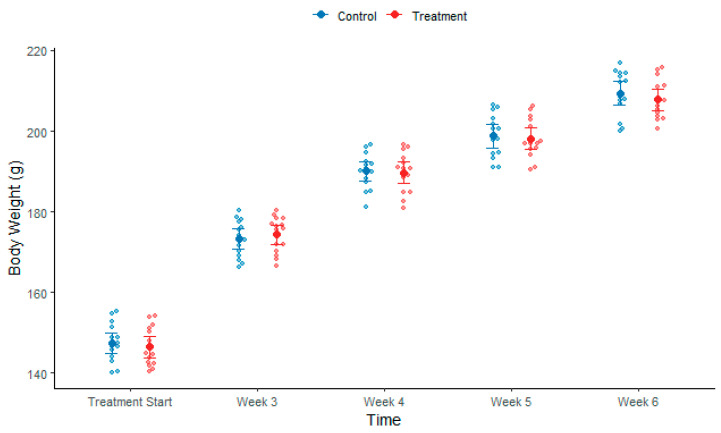
Bee swarm plot showing a gradual recovery in body weight during treatment phase both in control and treatment groups from start of the treatment until week 6.

**Figure 6 biomolecules-15-01647-f006:**
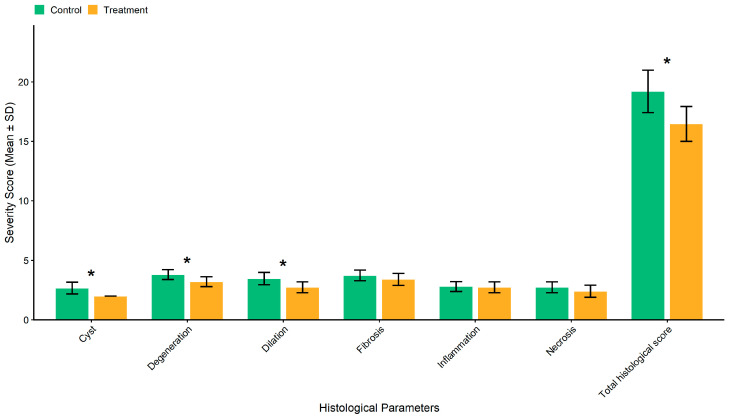
Histological analysis (Renal degeneration, cyst formation, and tubular dilation show significant improvement). * Shows a statistically significant difference (*p* < 0.001) compared to the Control group.

**Figure 7 biomolecules-15-01647-f007:**
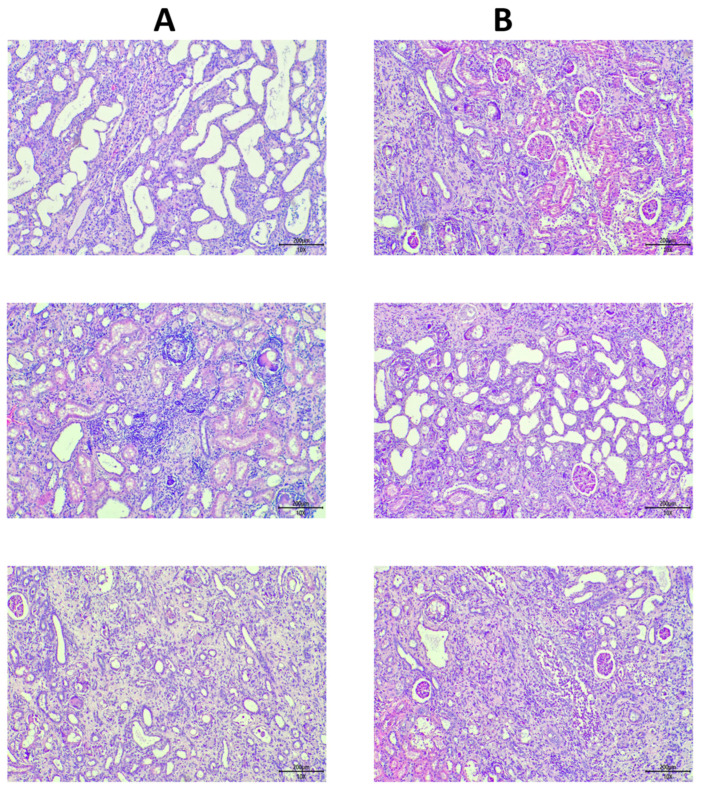
Representative histological sections showing (**A**) Control and (**B**) Treatment Group. The exosome treated group demonstrates significant improvement in renal degeneration, cyst formation, and tubular Dilation.

**Table 1 biomolecules-15-01647-t001:** Criteria of severity grading system for histopathological parameters.

Grade	Severity
1	Minimal (<10%)
2	Mild (11–39%)
3	Moderate (40–79%)
4	Marked (80–100%)

**Table 2 biomolecules-15-01647-t002:** Serum creatinine and urinary protein levels before chronic kidney disease (CKD) induction. Both groups started at the comparable baseline.

**Feline Exosome Treated Group**															
**Animal ID**	1001	1002	1003	1004	1005	1006	1007	1008	1009	1010	1011	1012	1013	1014	1015
**Creatinine (mg/dL)**	0.6	0.5	0.5	0.6	0.6	0.5	0.6	0.7	0.5	0.6	0.7	0.5	0.6	0.7	0.5
**Urinary Protein (mg/dL)**	-	-	-	-	-	-	-	-	-	-	-	-	-	-	-
**Control Group**															
**Animal ID**	1016	1017	1018	1019	1020	1021	1022	1023	1024	1025	1026	1027	1028	1029	1030
**Creatinine (mg/dL)**	0.5	0.6	0.6	0.7	0.7	0.5	0.6	0.5	0.5	0.7	0.6	0.6	0.5	0.5	0.7
**Urinary Protein (mg/dL)**	-	-	-	-	-	-	-	-	-	-	-	-	-	-	-

- means the protein was not detected.

**Table 3 biomolecules-15-01647-t003:** Serum creatinine and urinary protein levels after chronic kidney disease (CKD) induction. After induction, both serum creatinine and urinary protein levels are above the normal range.

**Feline Exosome Treated Group**															
**Animal ID**	1001	1002	1003	1004	1005	1006	1007	1008	1009	1010	1011	1012	1013	1014	1015
**Creatinine (mg/dL)**	1.8	1.4	1.6	1.5	1.3	1.3	1.6	1.8	1.5	1.4	1.3	1.8	1.4	1.6	1.7
**Urinary Protein (mg/dL)**	>30	>30	>30	>30	>30	>30	>30	>30	>30	>30	>30	>30	>30	>30	>30
**Control Group**															
**Animal ID**	1016	1017	1018	1019	1020	1021	1022	1023	1024	1025	1026	1027	1028	1029	1030
**Creatinine (mg/dL)**	1.7	1.6	1.4	1.5	1.3	1.6	1.4	1.8	1.7	1.3	1.3	1.7	1.5	1.8	1.4
**Urinary Protein (mg/dL)**	>30	>30	>30	>30	>30	>30	>30	>30	>30	>30	>30	>30	>30	>30	>30

**Table 4 biomolecules-15-01647-t004:** Serum creatinine and urinary protein levels after treatment.

**Feline Exosome Treated Group**															
**Animal ID**	1001	1002	1003	1004	1005	1006	1007	1008	1009	1010	1011	1012	1013	1014	1015
**Creatinine (mg/dL)**	1	0.9	0.8	0.8	0.9	1	1.1	0.9	0.8	0.8	0.9	1	0.8	0.8	0.8
**Urinary Protein (mg/dL)**	-	-	-	-	-	-	-	-	-	-	-	-	-	-	-
**Control Group**															
**Animal ID**	1016	1017	1018	1019	1020	1021	1022	1023	1024	1025	1026	1027	1028	1029	1030
**Creatinine (mg/dL)**	0.9	0.9	1	0.8	0.9	0.8	0.9	1	1.1	1	1	0.9	0.8	0.9	0.9
**Urinary Protein (mg/dL)**	-	-	-	-	-	-	-	-	-	-	-	-	-	-	-

- means the protein was not detected.

## Data Availability

The raw data supporting the conclusions of this article will be made available by the authors on request.

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
