# Peer review of "Evaluation of Feline Exosome Mediated Renal Regeneration in Adenine-Induced Chronic Kidney Disease"

_biomolecules, 2025, doi:10.3390/biom15121647_

Round 1
Reviewer 1 Report
Comments and Suggestions for Authors
The subject is interesting and the regenerative potential of exosomes in kidney damage opens new strategies in delaying CKD progression. However, several issues concerning this study have to be raised.
After induction of CKD, both groups (control group and treatment group) presented with decreasing weight, rising creatinine and proteinuria. After 4-week treatment with exosomes, both groups showed improvement in all above mentioned parameters, though none of them reached pre-induction values. Having said that, regression was spontaneous and independent of treatment.
CKD by definition is progressive and irreversible, so the desired effect would rather be to slow down the progression or to stop it. If the results improved, are we still within the CKD category, or maybe these anomalies were acute in their character, and we faced either AKI or AKD, not CKD? Please comment on that.
Indeed, histopathological reversal of some aspects of damage sound promising, and probably improvement of kidney function parameters would follow if the observation time took longer – a suggestion for future studies.
Graphical presentation of serum creatinine values should be compressed to one figure (fig. 4 and 6 compiled). Analogous data concerning proteinuria should be added.
Discussion is rather short. The essential part should contain the comparison of current results with other studies. Ref. 23 is a review, so there should be more focus on experimental data extracted from this article (ref. 67/68 cited in ref.23).
Author Response
We thank you for your time to review and improve our manuscript. Please see the attachment for our point-by-point response.

Reviewer 2 Report
Comments and Suggestions for Authors
Title: Evaluation of Feline Exosome Mediated Renal Regeneration in an Adenine-Induced Chronic Kidney Disease
The topic is interesting and timely. However, the current manuscript does not provide sufficient experimental rigor to support the conclusion of “renal regeneration.” Structural differences in histology alone, without validated functional or mechanistic markers, are not enough to claim therapeutic benefit. The major problem is that the study design and results do not support the conclusion. The conclusions claim structural benefit from exosomes, but the model, treatment duration, dose rationale, functional biomarkers, and histopathology quantification are weak or insufficient to justify “renal regeneration”.
Major Comments
- In the Results part, Only H&E and Masson are presented. No molecular markers were checked (no Ki67, no KIM-1, no NGAL, no α-SMA, no collagen-I, no TGF-β).
- Creatinine was measured using test strips. This is not acceptable for a peer-reviewed biomolecular journal. Creatinine should be quantified by autoanalyzer (IDEXX alone is not gold standard) or enzymatic assay (serum chemistry). Also no BUN, no eGFR (rat), no urine microalbumin, no cystatin-C. Functional effect cannot be judged.
- Exosome dosing is unclear / not justified. Dose = 0.1 mL per rat. But how many particles was this? (particles/animal); what was protein amount? ; what was concentration after reconstitution?; Dose should be expressed as particles / kg or µg protein / kg.
- No exosome characterization shown in this data. NTA alone is not enough. MSC-derived exosomes must show at least: CD63 / CD81 / CD9 (Western blot or ExoView), TEM morphology image
- Adenine CKD induction is too short and too mild. Only 2 weeks adenine is insufficient to generate stable chronic fibrosis. Standard adenine CKD models use 4+ weeks to establish persistent fibrosis and chronic impairment. 2 weeks adenine + 4 weeks recovery → kidney spontaneously improves. So the “improvement” could be natural recovery, not exosomes.
- Statistics do not match small effect size: n=15/group is good, but results show no functional difference Claiming structural regeneration with no functional benefit is speculative.
- The text constantly says “regenerative effect”. But data do NOT support regeneration. At best: “possible structural amelioration in tubular injury parameters without functional recovery.”
MinorComments
- English needs substantial revision—tone is repetitive, too long, and lacks clarity.
- Introduction is too long; shorten global CKD statistics and focus more on rationale.
- Figures 7 & 8 are very poor quality, unclear resolution. Labeling is insufficient.
- Table 1 severity scale is inconsistent (Grade 3 says 40–79%, huge range; typo?).

Author Response

(The authors gave the same response as above.)

Round 2
Reviewer 1 Report
Comments and Suggestions for Authors
All concerns have been addressed - I have no further comments
Reviewer 2 Report
Comments and Suggestions for Authors
This paper clearly revised according to the reviewer's comments.